# Atomic stiffness for bulk modulus prediction and high-throughput screening of ultraincompressible crystals

Ruihua Jin[1,2], Xiaoang Yuan[1,2] & Enlai Gao [1] ✉

Determining bulk moduli is central to high-throughput screening of ultra-incompressible materials. However, existing approaches are either too inaccurate or too expensive for general applications, or they are limited to narrow chemistries. Here we define a microscopic quantity to measure the atomic stiffness for each element in the periodic table. Based on this quantity, we derive an analytic formula for bulk modulus prediction. By analyzing numerous crystals from first-principles calculations, this formula shows superior accuracy, efficiency, universality, and interpretability compared to previous empirical/semiempirical formulae and machine learning models. Directed by our formula predictions and verified by first-principles calculations, 47 ultra-incompressible crystals rivaling diamond are identified from over one million material candidates, which extends the family of known ultraincompressible crystals. Finally, treasure maps of possible elemental combinations for ultra-incompressible crystals are created from our theory. This theory and insights provide guidelines for designing and discovering ultraincompressible crystals of the future.

Bulk modulus is an important quantity of condensed matters, which not only measures volumetric elasticity, but also closely correlates to other non-elastic properties, such as hardness and toughness[1]. Driven by demands for promising uses competing with the diamond in industrial applications, such as cutting tools, and high-pressure devices, much effort has been devoted to searching for ultra-incompressible, superhard materials[2–6]. The term "ultraincompressible" has been diversely used for materials with a bulk modulus higher than 300 GPa, 350 GPa, or 400 GPa[7–9]. In this work, we define that a material is ultraincompressible if its bulk modulus is higher than 400 GPa, which can rival diamond (436 GPa). To discover ultraincompressible materials from a vast number of candidates, several approaches for the prediction of bulk moduli have been developed, including empirical/semi-empirical formulae[10–12], first-principles calculations[2,3,13,14], and machine learning predictions[5,15,16]. From these approaches, achievements have been made in the past decades and a few ultraincompressible materials with a bulk modulus rivaling diamond, such as $C_3N_4$ (425-496 GPa)[2], BN

(400 GPa)[17,18], ReC (422 GPa)[19] and lonsdaleite (438 GPa)[20,21], have been reported.

There is still plenty of room to discover ultraincompressible materials, since the energy landscape is exceedingly complex because of infinite elemental combinations. However, as crystal structures identified from experiments and calculations in databases [e.g., Crystallography Open Database (COD)[22], Inorganic Crystal Structure Database (ICSD)[23], Materials Project (MP)[24], and Open Quantum Materials Database (OQMD)[25]] increase rapidly, it becomes challenging to determine bulk moduli using the above-mentioned approaches because of requirements for accuracy, efficiency, and universality. First-principles calculation is the most universal and accurate approach. However, after years of effort, only less than 1% of crystals (~$10^4$) have first-principles-calculated bulk moduli, while the total number of crystals is more than $10^6$ in these databases. It means that the bulk moduli for more than 99% of crystals in these databases remain unknown. This is because high-throughput large-scale first-principles stress-strain or phonon

[1]Department of Engineering Mechanics, Wuhan University, Wuhan, Hubei 430072, China. [2]These authors contributed equally: Ruihua Jin, Xiaoang Yuan.
✉e-mail: enlaigao@whu.edu.cn

calculations of elastic constants are too costly and time-consuming, while previous empirical/semiempirical formulae are only applicable to a narrow range of materials. Hence, the high cost of first-principles calculations and weak universality of previous empirical/semiempirical formulae limit their uses for high-throughput prediction of bulk moduli for a vast number of diverse candidates ($>10^6$). To accelerate this process, machine learning approaches have demonstrated immense potential in recent years[5,16]. Despite achievements made, the identified ultraincompressible materials with a bulk modulus higher than 400 GPa directed by machine learning approaches are still very few. For example, machine learning-directed searching of 18,493 materials only found one material with a bulk modulus higher than 400 GPa (401 GPa for $Re_2C$)[16]. This is because there are critical issues, such as a lack of suitable descriptors[26], poor interpretability, and insufficient training in the range of high-moduli, which limit predictive accuracy. Therefore, predicting bulk moduli with high accuracy, high efficiency, strong universality, and high interpretability is needed.

In this work, we propose a microscopic quantity to measure atomic stiffness for each element within the periodic table. Based on this quantity, an analytic formula is derived for predicting bulk moduli of crystals. Compared to previous empirical/semiempirical formulae and machine learning predictions, this theoretical formula shows the advantages of high accuracy, high efficiency, strong universality, and high interpretability. Directed by the prediction from our formula and verified by first-principles calculations, we identify 47 ultra-incompressible crystals with a bulk modulus rivaling diamond from over one million material candidates. To guide the discovery and design of ultraincompressible materials of the future, possible elemental combinations for ultraincompressible materials are predicted and discussed.

## Results

### Theory underlying the prediction of bulk moduli from atomic stiffnesses

The bulk modulus of a material is defined as the ratio of the infinitesimal pressure increase to the resulting relative decrease of the volume (Fig. 1a). At the atomic scale, the interatomic interaction can be looked upon as the atomic energy (energy per atom, $E$) that is a function of the atomic volume (volume per atom, $V$). From the definition of bulk modulus (i.e., $B = V d^2E/dV^2$), it can be derived that

$$BV = d^2E/d\varepsilon^2|_{\varepsilon \to 0}, \tag{1}$$

where $\varepsilon$ is the volumetric strain. The bulk modulus is related to the atomic energy gradient ($d^2E/d\varepsilon^2|_{\varepsilon \to 0}$) on the limit of small volumetric strain (Fig. 1b). By considering that the atomic energy of a compound ($E$) is contributed from each atom therein, and assuming that the volume deformation of compounds is affine ($\varepsilon = \varepsilon_i$), it can be further obtained that

$$d^2E/d\varepsilon^2 = \sum x_i d^2E_i/d\varepsilon_i^2, \text{ i.e., } BV = \sum x_i B_i V_i, \tag{2}$$

where $x_i$, $E_i$, $\varepsilon_i$, $B_i$, and $V_i$ are the atomic fraction, atomic energy, volumetric strain, bulk modulus, and atomic volume for element i in this compound, respectively (Fig. 1c). Subsequently, the bulk modulus of this compound can be derived as

$$B = \sum x_i B_i V_i / V. \tag{3}$$

This volume-weighted mean is similar to the classical rule of mixtures for composites[27]. For example, the elastic modulus of fiber-reinforced composites can be estimated by the rule of mixtures as $E_c = xE_f + (1-x)E_m$, where $x$ is the volume fraction of the fibers, and $E_f$ and $E_m$ are the elastic moduli of the fibers and the matrix, respectively. The difference is that, compared to volume conservation in the classical rule of mixtures for composites, $V$ for a compound does not have to be equal to $\sum x_i V_i$, since significant volume change might occur from components in the pure form (simple substances) to compounds.

Equation 3 indicates that the bulk modulus for a compound can be predicted if the product of bulk modulus and atomic volume ($B_i V_i$) for each element therein is determined, in which the atomic fraction of each element ($x_i$) can be obtained from the chemical formula of the compound. $B_i$ is a macroscopic quantity for a material, while $V_i$ is a microscopic quantity for each atom therein. Their product is equal to

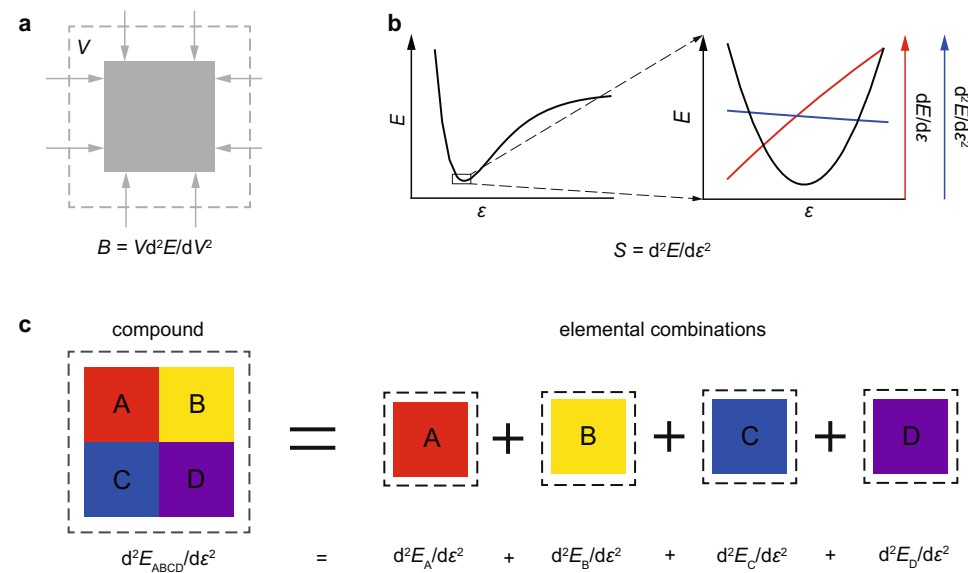

**Fig. 1 | Illustration of the theory underlying the prediction of bulk moduli from atomic stiffnesses. a** Illustration of the bulk modulus, where $B$ and $V$ are the bulk modulus and atomic volume of a material, respectively. **b** Illustration of atomic energy, and first and second derivatives of atomic energy with respect to the volumetric strain, where $S$, $E$, and $\varepsilon$ are the atomic stiffness, atomic energy, and volumetric strain, respectively. **c** Illustration of a compound consisting of atoms A, B, C, and D, whose atomic energy gradient equals the sum of the atomic energy gradient of each atom therein by assuming that the volume deformation of the compound is affine.

the atomic energy gradient in the equilibrium state ($B_i V_i = d^2 E_i / d\varepsilon_i^2 |_{\varepsilon_i \to 0}$), which describes the resistance of an atom to volumetric strain. A literature survey shows that such quantity that characterizes atomic stiffnesses has not been defined before. Hence, we here defined this microscopic quantity ($S_i = B_i V_i$) as "atomic stiffness", which has the same dimension as the well-known atomic cohesive energy. From atomic stiffnesses, the bulk modulus can be calculated by

$$B = \sum x_i S_i / V. \qquad (4)$$

It is known that the bulk modulus originates from the resistance of chemical bonds to compression. The parameters of chemical bonds are implicitly included in this analytic formula, since it has been demonstrated that the proposed atomic stiffness of each atom ($S_i$) is related to the coordination number ($N$), the bond force constant ($k$), and the bond length ($d$) of chemical bonds around the atom (see Methods for details). It should be noted that the parametrizations of bond force constant, bond length, and coordination number are challenging, since they depend on local chemical environments. We circumvented these challenging parametrizations by using the concept of atomic stiffness that is insensitive to local chemical environments as demonstrated below.

### Determining the atomic stiffness for each element within the periodic table

To predict bulk moduli from the atomic stiffnesses (Eq. 4), we determined the atomic stiffness ($S_i$) for each element within the periodic table (Fig. 2a). It has been known that an element can form more than one crystalline form, i.e., allotropes. Interestingly, although different allotropes of the same element have very different bulk moduli and atomic volumes, their atomic stiffnesses are close (Supplementary Data 1). For example, although the bulk moduli for diamond (436 GPa) and graphite (275 GPa) differ significantly, the atomic stiffnesses for diamond (15.55 eV) and graphite (15.70 eV) only have a difference of less than 1%. The insensitivities of atomic stiffness values for different allotropes can be understood from the following analyses: (1) It has been reported that the bulk modulus ($B_i$) has a relation with the bond length ($d$) as $B_i \sim d^n$, where n is approximately $-3$[10,28], while the atomic volume ($V_i$) has a relation with the bond length as $V_i \sim d^3$. Hence, the atomic stiffnesses ($S_i = B_i V_i$) of different allotropes for each element are almost a constant. (2) It has been reported that the bulk modulus ($B_i$) is proportional to the energy density ($\rho_i = E_i / V_i$, where $E_i$ is the atomic energy): $B_i \sim E_i / V_i^2$[29], i.e., $B_i V_i = S_i \sim E_i$. Because allotropes would be unstable if their energies are significantly higher than the most stable phase, the atomic energies of different allotropes ($E_i$) are usually very close (e.g., 7.35 eV of diamond and 7.37 eV of graphite)[30], resulting in similar atomic stiffnesses ($S_i$) for different allotropes of each element. Moreover, the atomic stiffness changes slightly for a certain range of volumetric strain (Supplementary Fig. 1), exhibiting a strain-insensitive feature. These insensitivities of atomic stiffness to local chemical environments are particularly beneficial to the simple yet universal prediction of bulk moduli from atomic stiffnesses, since we only need to determine and use one value of atomic stiffness for each element.

The determination of atomic stiffness for each element within the periodic table is based on the following rules: (1) For most elements whose simple substances are solids at standard temperature and pressure, we here used the well-known dense crystalline form of simple substances to parametrize the atomic stiffnesses by considering that the state of each element in ultraincompressible compounds is close to their dense crystalline form. The dense crystalline forms of simple substances were extracted from the MP database[24]. The atomic stiffness ($S_i$) for each element was calculated from the bulk modulus ($B_i$) and atomic volume ($V_i$) of the corresponding structure. (2) For a few elements whose simple substances are gases, liquids, or weakly

interacting molecular crystals at standard temperature and pressure (e.g., N and O), the atomic stiffnesses were derived from their compounds in MP database that have first-principles-calculated bulk moduli (see Methods for details). Subsequently, $B_i$, $V_i$, and $S_i$ for elements within the periodic table were parametrized (Fig. 2a). It should be noted that hydrogen, noble gas, and radioactive elements were omitted, since they are unlikely to form high-bulk-modulus compounds with any other elements.

These bulk moduli, atomic volumes, and atomic stiffnesses exhibit periodic dependence upon the atomic number (Fig. 2b). This is because atomic properties of the elements, such as atomic volume, atomic valance electron, show periodic dependence upon the atomic number[1]. Meanwhile, the bulk moduli also exhibit periodic dependence upon the atomic number, since it is related to the valance electron density, i.e., atomic valance electron divided by atomic volume[1]. Consequently, as the product of bulk modulus and atomic volume, the atomic stiffness also shows periodic dependence upon the atomic number. With the table of parametrized atomic stiffness ($S_i$) for each element, the bulk moduli for compounds can be calculated by Eq. 4.

### Evaluating the predictive performance of bulk moduli from atomic stiffnesses

To evaluate the predictive performance of our formula, we did comparisons with previous approaches (i.e., empirical/semiempirical formulae, machine learning models, and first-principles calculations). Among these approaches, first-principles calculation is the most trustworthy yet high-cost approach for predicting bulk moduli. Hence, the bulk moduli for 6192 crystals in the MP database that have a first-principles-calculated bulk modulus after data cleaning were calculated by other approaches (Cohen's formula[14], Li's formula[11], our formula), and the mean absolute relative errors (MAREs) and mean absolute errors (MAEs) of bulk moduli predicted from our formula ($B_O$), Cohen's formula ($B_{Cohen}$)[14], and Li's formula ($B_{Li}$)[11] were calculated with respect to first-principles calculations ($B_F$) (see Methods for details).

First, we compared our formula with previous empirical/semiempirical formulae (Cohen's formula[14], Li's formula[11]). It should be noted that Cohen's formula[14] and Li's formula[11] are only applicable to limited types of crystals (e.g., diamondlike and zinc-blende compounds). As these previous empirical/semiempirical formulae were artificially extended to all 6192 crystals (see Methods for details), the MAREs of $B_{Cohen}$, $B_{Li}$, and $B_O$ with respect to $B_F$ are 37.7%, 42.1%, and 19.6%, respectively (Fig. 3). These results indicate that previous empirical/semiempirical formulae are only accurate for a narrow range of crystals, and their accuracies would break down as extended to numerous crystals having diverse structures and compositions, and that our formula is substantially more universal and more accurate than previous empirical/semiempirical formulae.

Next, we compared our formula with recent machine learning models[5,16]. Despite achievements made from machine learning, the discovered ultraincompressible materials directed by machine learning models are still very few. For example, machine learning-directed searching of 18,493 materials only identified one ultraincompressible crystal ($Re_2C$ with a bulk modulus of 401 GPa)[16]. We here did a comparison with two literature-reported machine learning models for predicting elastic moduli[5,16]. The first machine learning model having 55 descriptors was developed based on a deep neural network[16]. The raw data of the top 50 compounds with the largest predicted bulk moduli was provided, which can be used for a direct comparison. For these top 50 compounds, the MAREs of the machine learning model and our formula are 4.8% and 5.1%, respectively (Supplementary Data 2). The second machine learning model having 150 descriptors was developed based on a support vector machine regression[5]. It shows that the MARE of machine learning predicted bulk moduli with respect to first-principles calculations is about 15%, and the bulk

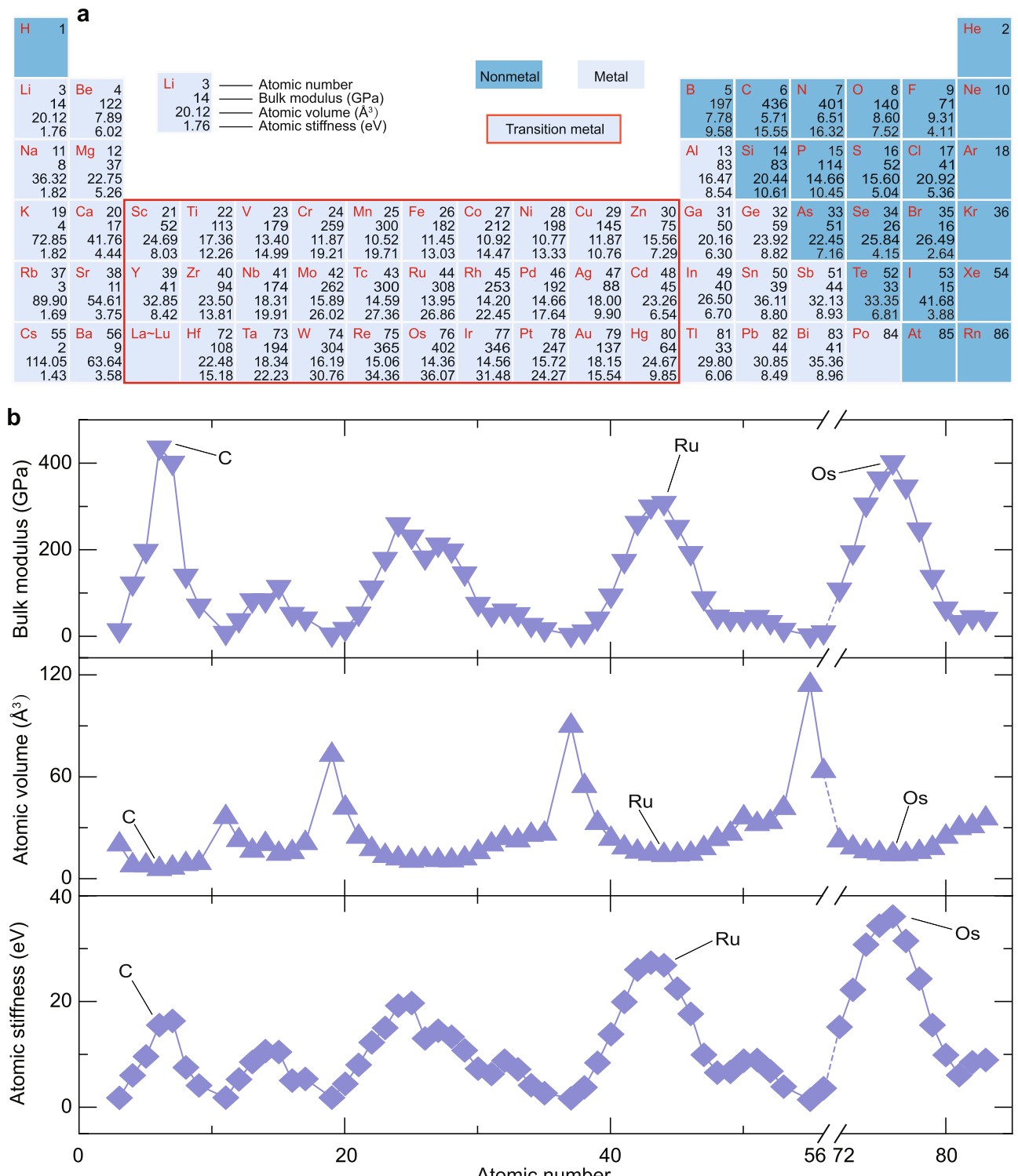

**Fig. 2 | Atomic parameters for most elements in the periodic table. a** Bulk moduli, atomic volumes, and atomic stiffnesses for elements within the periodic table. **b** Periodic dependences of bulk moduli, atomic volumes, and atomic stiffnesses upon the atomic number. Source data are provided as a Source Data file.

moduli are underestimated in the range of high bulk moduli. As a comparison, the MARE of our formula for crystals with the bulk moduli around 200 GPa and 400 GPa are 13.5% and 5.6%, respectively, indicating increasingly accurate prediction as the bulk moduli increase (Fig. 3a). The more accurate prediction for higher-bulk-modulus crystals might be attributed to that the assumption of affine deformation made in our theory is better satisfied for such crystals. This is very useful in the following high-throughput screening, since crystals

with a high bulk modulus are just what we are looking for. These two comparisons indicate that the accuracy of our formula is comparable to and even higher than previous machine learning models. In addition to comparable and even higher accuracy, our theoretical formula has clear advantages of high interpretability, high efficiency, and great ease of use as compared with recent machine learning models.

Additionally, it can be found that the bulk moduli of a few crystals predicted from our formula ($B_O$) and first-principles calculations in the

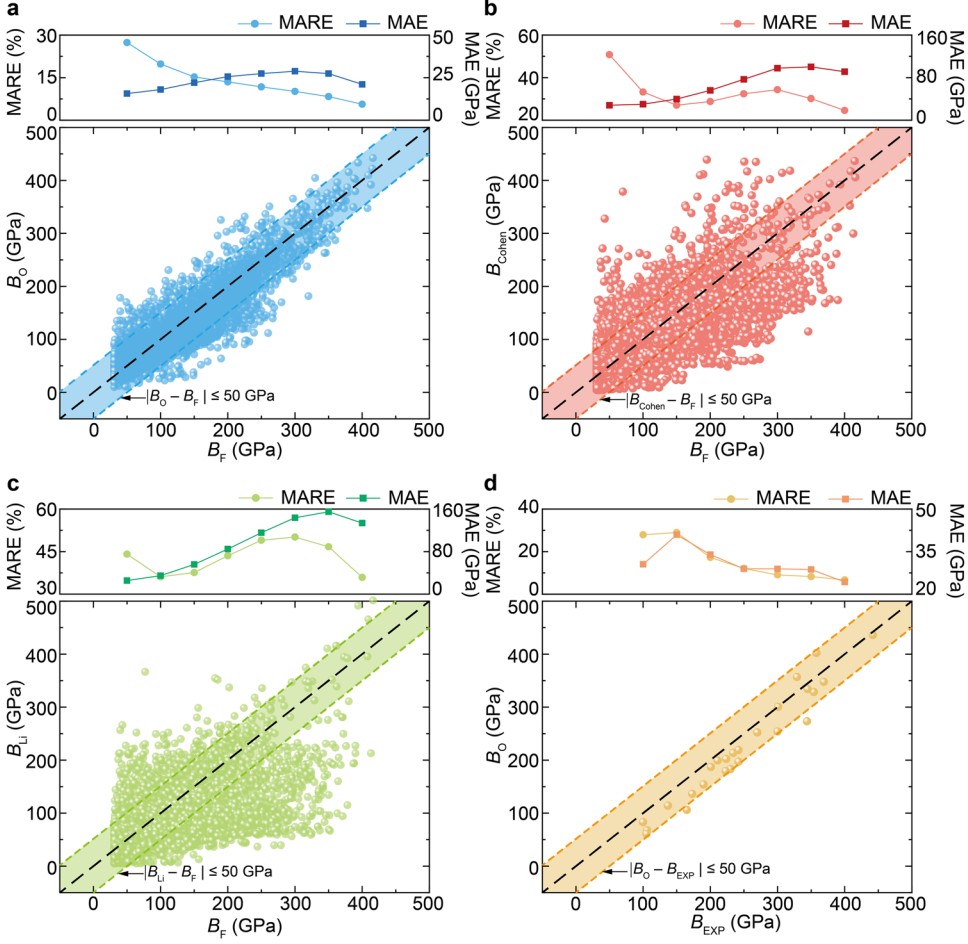

**Fig. 3 | Comparison of bulk moduli predicted from our formula ($B_O$), previous empirical/semiempirical formulae ($B_{Cohen}$[14] and $B_{Li}$[11]) with first-principles calculations ($B_F$), and comparison of $B_O$ with experimental measurements ($B_{EXP}$).** Predicted bulk moduli of crystals in Materials Project (MP) database from **a** our formula, **b** Cohen's formula, and **c** Li's formula, as compared with first-principles calculations from MP database ($B_F$). **d** Comparison of $B_O$ and $B_{EXP}$. Each point of mean absolute relative error (MARE) and mean absolute error (MAE) is counted for crystals within the range of ±50 GPa. The shadows represent the region where the deviation of the bulk modulus is less than 50 GPa. Source data are provided as a Source Data file.

MP database ($B_F$) have very large differences (Fig. 3a). To trace the origin of the large deviations, we reperformed high-fidelity first-principles calculations of bulk moduli ($B_{HF}$) by adopting more stringent convergence criteria than that used in MP database (see Methods for details). First, we selected 16 crystals having a large deviation ($|B_O − B_F| > 80$ GPa) and reperformed high-fidelity first-principles calculations (Supplementary Fig. 2a). The MAREs of $B_O$ and $B_F$ in the MP database with respect to $B_{HF}$ for these 16 crystals are 20.8% and 103.7%, respectively (Supplementary Table 1). Moreover, we randomly selected 306 crystals from the above 6192 crystals and reperformed high-fidelity first-principles calculations of bulk moduli. The MARE of $B_O$ with respect to $B_F$ is 26.4%, while the MARE of $B_O$ with respect to $B_{HF}$ is 17.1% (Supplementary Fig. 2b). It can be found that $B_F$ and $B_{HF}$ for a few crystals also have a very large difference, while $B_O$ is generally consistent with $B_{HF}$. These results indicate that the large difference between $B_O$ and $B_F$ for a few crystals might result from calculation issues (e.g., convergence criteria of first-principles calculations used in the MP database) or processing errors in the MP database, which have been also reported in previous work[5,15]. For these crystals, the first-principles moduli were manually corrected by our high-fidelity first-principles calculations. Finally, the efficiency of our formula was also investigated. The ratio of computational durations between first-principles calculations and our formula is on the order of $10^7$. These results indicate that our formula is so accurate that it can even be used

to correct the errors of regular first-principles calculations in material databases, while it is orders of magnitude faster than first-principles calculations. Finally, we did a literature survey and collected materials with experimentally measured bulk moduli[31–47] (Supplementary Table 2). The predicted bulk moduli of these materials by our formula are well consistent with experimental values (Fig. 3d), which provides experimental support for our theoretical prediction.

## Theory-directed search for ultraincompressible materials

We here predicted the bulk moduli for a vast number of materials whose bulk moduli remain unknown in material databases. To our knowledge, more than one million crystal structures have been collected in material databases to date, but only less than 1% of the crystals in material databases have a first-principles modulus, and only about 10% of them have been used in the past searching for ultraincompressible materials[5,16]. To extend the family of candidates, we collected a vast number of crystal structures ($> 10^6$) from two of the largest databases, MP and OQMD. After data cleaning, 512419 compounds were retained (see Methods for details). By substituting the values of $x_i$, $S_i$, and $V$ into the formula of $B = \sum x_i S_i / V$, the bulk moduli for these compounds were predicted, and 427 compounds have a predicted bulk modulus higher than 350 GPa. After removing duplicates, the first-principles calculations were performed for these compounds. Finally, 47 crystals with a bulk modulus higher than 400 GPa rivaling

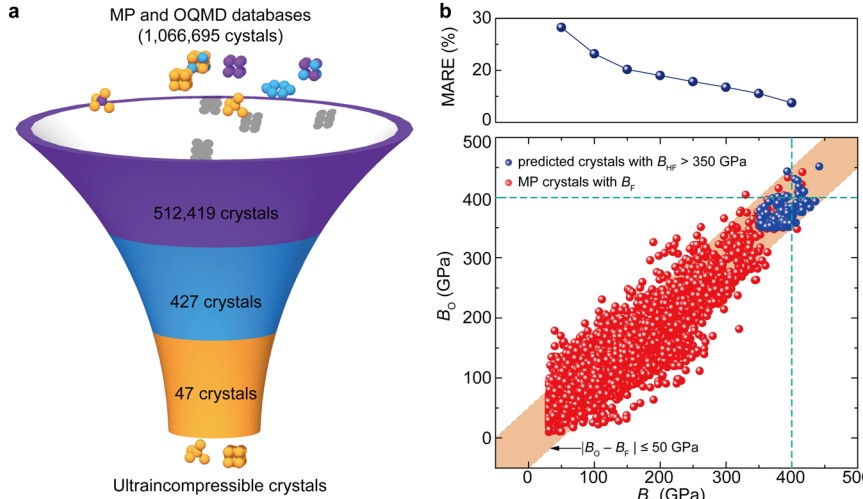

**Fig. 4 | High-throughput screening of ultraincompressible crystals.**
**a** Illustration of the screening process for ultraincompressible crystals from a vast number of crystals in Materials Project (MP) and Open Quantum Materials Database (OQMD). **b** Predicted bulk moduli from our formula ($B_O$) for 6192 crystals in the MP database that have a first-principles-calculated bulk modulus ($B_F$), and 240 crystals with a first-principles-calculated bulk modulus ($B_{HF}$) higher than 350 GPa in MP and OQMD databases. Each point of mean absolute relative error (MARE) is counted for crystals within the range of ±50 GPa. The shadow represents the region where the deviation of the bulk modulus is less than 50 GPa. The dashed line denotes the threshold of bulk modulus for incompressible materials defined in this work (400 GPa). Source data are provided as a Source Data file.

diamond (436 GPa) were identified from these material candidates (Fig. 4, the atomic coordinates of the optimized structures for these ultraincompressible crystals were provided in Supplementary Data 3). A literature survey shows that our search dramatically extends the family of ultraincompressible crystals (Table 1). These results indicate the power of our formula for high-throughput screening of ultra-incompressible materials.

## Discussion

Considering that elemental combinations are infinite, and the number of new materials is still growing rapidly, the next interesting questions might be (1) what the ultraincompressible materials of the future are? (2) how to efficiently design and discover the ultra-incompressible materials of the future? To answer these questions, a treasure map of elemental combinations for ultraincompressible materials would be useful. We found that the total volume of compounds with a high bulk modulus is approximately equal to the sum of the atomic volume of each atom in the dense crystalline form of the corresponding simple substance (Supplementary Fig. 3). By substituting the parameterized atomic stiffness and atomic volume of each element into our formula, the bulk moduli of these compounds can be predicted solely based on their chemical composition. Herein, we presented an example using binary compounds composed of two elements in equal atomic fractions (Fig. 5a). It can be found that the most possible elemental combinations for ultra-incompressible materials are C, C/N, C/Os, Os, N/Os, etc. (Fig. 5b), indicating that light main group elements (C, N, etc.), transition metal elements (Re, Os, Ir, etc.) and their compounds are the most promising candidates for ultraincompressible materials. This finding is consistent with empirical design rules for ultraincompressible, super-hard materials[48]. More specifically, previous design principles indicate that potential super-hard, ultraincompressible materials include (1) compounds composed of light elements (B, C, N, O, etc.) from periods 2 and 3 of the periodic table, as these elements can form short covalent bonds, and (2) compounds combining heavy transition metal elements with light elements, since the heavy transition metals contribute high valence electron density to the compounds, which enhances their resistance to mechanical deformation[48–50]. It should be noted that these previous design principles are mainly empirical and qualitative, while our formula offers a theoretical and quantitative approach (Fig. 5). As a result, our studies provide more accurate predictions and insights for designing ultraincompressible materials (bulk modulus > 400 GPa) compared to previous design principles: (1) The range of elements that can potentially form ultraincompressible materials is significantly narrowed. For instance, among light elements, carbon (C) and nitrogen (N) are the most promising candidates, while boron (B) and oxygen (O) are less likely to form ultraincompressible materials (bulk modulus > 400 GPa) when their atomic fractions are larger than 50%. (2) Alloys composed of heavy transition metal elements (Re, Os, Ir, etc.) have the potential to be ultraincompressible. (3) Once the chemical composition is determined in the search for ultraincompressible materials, efforts should be directed towards identifying structures with small atomic volumes (as indicated by Eq. 4). These treasure maps and insights provide guidelines for designing and discovering ultraincompressible materials of the future.

To summarize, we define a microscopic quantity, i.e., atomic stiffness, for each element in the periodic table to characterize the resistance of an atom to volumetric strain. From the atomic stiffnesses, we further derive a general formula for predicting the bulk moduli of crystals. By analyzing a large number of crystals from first-principles calculations, this formula shows the all-in-one advantage of high accuracy, high efficiency, strong universality, and high interpretability compared to previously proposed approaches. Directed by this formula prediction and verified by first-principles calculations, 47 ultra-incompressible crystals have been identified, extending the family of known ultraincompressible crystals. Finally, treasure maps of possible elemental combinations for ultraincompressible materials of the future are created and discussed.

## Methods

### Correlation between the atomic stiffness and the chemical bonding parameters
Taking the diamond as an example, the atomic energy of an atom is contributed by the energy of the chemical bond around the atom:

$$E = NE_b/2, \quad (5)$$

**Table 1 | Identified ultraincompressible crystals**

| ID | Formula | $B_O$ (GPa) | $B_F$ (GPa) | $B_L$ (GPa) | Space group |
|---|---|---|---|---|---|
| OQMD-14925 | $C_3N_4$ | 452 | 442 | 496[2] | I-43d |
| OQMD-21515 | ReC | 394 | 436 | 436-457[55] | P-6m2 |
| OQMD-22207 | $Re_2C$ | 384 | 428 | 389[16] | P63/mmc |
| MP-867141 | $ReOs_3$ | 393 | 425 | 390[16] | P63/mmc |
| OQMD-1625530* | $ReIrOs_6$ | 394 | 422 | | P-6m2 |
| OQMD-22377 | $Re_2N$ | 393 | 422 | 411[56] | P63/mmc |
| OQMD-22307 | ReN | 410 | 419 | 453[57] | P63/mmc |
| OQMD-319530* | $IrOs_3$ | 394 | 419 | | P63/mmc |
| OQMD-1625527* | $MnIrOs_6$ | 385 | 418 | | P-6m2 |
| OQMD-347040* | $ReOs_3$ | 395 | 417 | | Pm-3m |
| OQMD-22378 | $Re_3N$ | 386 | 417 | 400[56] | P-6m2 |
| OQMD-1230026* | IrOs | 376 | 416 | | P-6m2 |
| OQMD-1435176* | $Re_3C$ | 379 | 416 | | P-6m2 |
| OQMD-301434* | $ReOs_3$ | 400 | 416 | | I4/mmm |
| OQMD-1625534* | $MnReOs_6$ | 387 | 415 | | P-6m2 |
| OQMD-1230666* | ReOs | 388 | 415 | | P-6m2 |
| OQMD-8183 | OsC | 403 | 414 | 396[58] | P-6m2 |
| OQMD-304348* | $IrOs_3$ | 394 | 414 | | I4/mmm |
| OQMD-326001* | IrOs | 376 | 413 | | R-3m |
| MP-1079494 | ReC | 388 | 413 | 411[59] | P63/mmc |
| OQMD-344016 | $IrOs_3$ | 390 | 413 | 421[60] | Pm-3m |
| OQMD-325825* | ReN | 408 | 411 | | R-3m |
| OQMD-14924 | $C_3N_4$ | 418 | 411 | 451[2] | P-3 |
| OQMD-22256 | OsC | 405 | 411 | 369[61] | P63/mmc |
| OQMD-337952* | ReOs | 388 | 410 | | P4/mmm |
| OQMD-1625528* | $IrOs_6W$ | 385 | 410 | | P-6m2 |
| OQMD-1625537* | $ReOs_6W$ | 388 | 409 | | P-6m2 |
| MP-1102681 | $CN_2$ | 428 | 409 | 407[62] | I42d |
| OQMD-327494* | ReOs | 387 | 409 | | R-3m |
| OQMD-326779* | ReC | 391 | 407 | | R-3m |
| MP-1219533 | ReIr | 358 | 406 | 337[16] | P6m2 |
| OQMD-319188* | $Os_3Ru$ | 386 | 406 | | P63/mmc |
| MP-1104073* | $C_{11}N_4$ | 431 | 405 | | P42m |
| MP-1186060* | $MnOs_3$ | 377 | 405 | | P63/mmc |
| OQMD-1625541* | $MoReOs_6$ | 383 | 405 | | P-6m2 |
| OQMD-1625539* | $CrReOs_6$ | 385 | 404 | | P-6m2 |
| OQMD-323983* | $TcOs_3$ | 383 | 403 | | P63/mmc |
| OQMD-336459* | IrOs | 377 | 403 | | P4/mmm |
| OQMD-1625529* | $CrIrOs_6$ | 382 | 403 | | P-6m2 |
| MP-867264 | $Re_3Os$ | 374 | 402 | 371[16] | P63/mmc |
| OQMD-1625533* | $MnCrOs_6$ | 377 | 402 | | P-6m2 |
| OQMD-301945* | $Os_3Pt$ | 370 | 402 | | I4/mmm |
| OQMD-327347* | ReIr | 360 | 402 | | R-3m |
| OQMD-1625866* | $Re_6IrOs$ | 373 | 401 | | P-6m2 |
| OQMD-301470* | $Re_3Os$ | 382 | 401 | | I4/mmm |
| OQMD-1625532* | $MnOs_6W$ | 377 | 401 | | P-6m2 |
| OQMD-345928* | $MnOs_3$ | 378 | 401 | | Pm-3m |

$B_O$, $B_F$, and $B_L$ denote the bulk moduli obtained from our formula prediction, first-principles calculations, and literature report, respectively. Newly found crystals are denoted by the symbol of *.
Source data are provided as a Source Data file.

where $N$ and $E_b$ are the coordination number and the energy of a chemical bond, respectively. Meanwhile, $E_b$ can be written as

$$E_b = kd^2\varepsilon_b^2/2, \tag{6}$$

where $k$, $d$, and $\varepsilon_b$ are the bond force constant, the equilibrium bond length, and the bond strain, respectively. Herein, the bond strain ($\varepsilon_b$) has a relation with the volumetric strain ($\varepsilon$) under a small strain: $\varepsilon = 3\varepsilon_b$. From the definition of atomic stiffness ($S_i = d^2E/d\varepsilon^2 |_{\varepsilon \to 0}$), the atomic

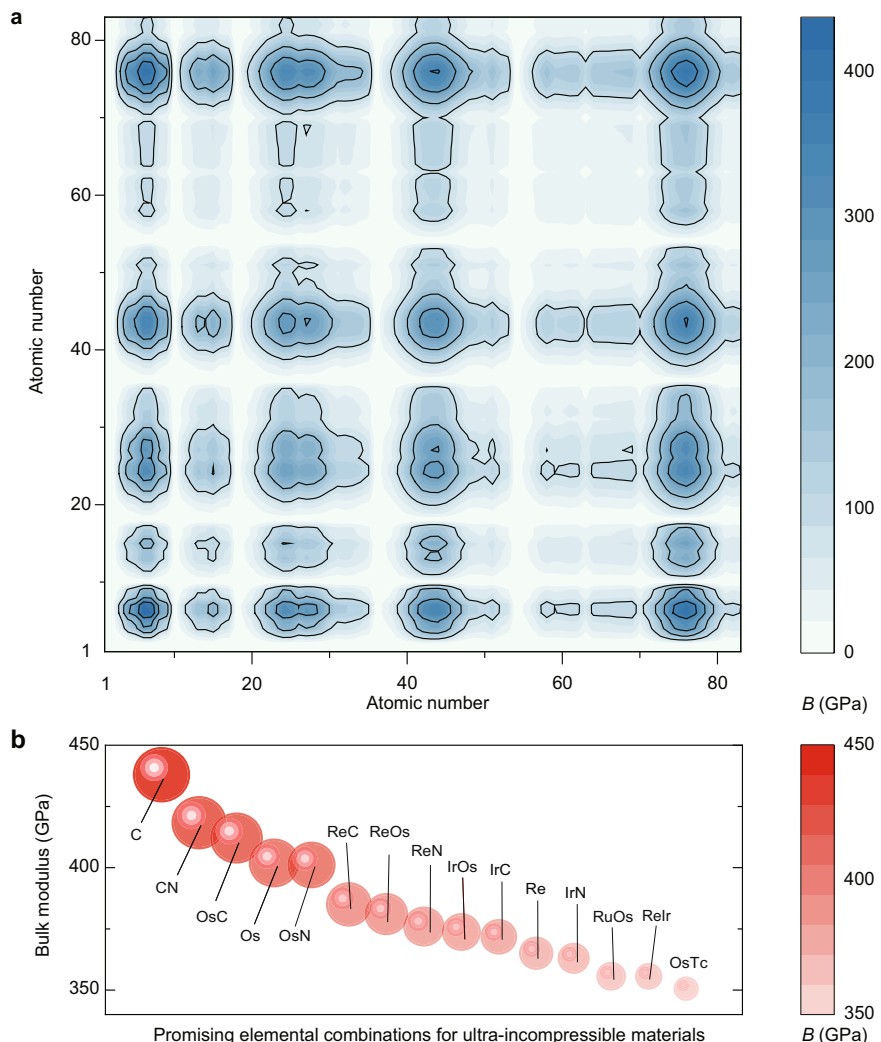

**Fig. 5 | Treasure maps of elemental combinations for ultraincompressible materials. a** Estimation of the bulk moduli for binary compounds composed of two elements in equal atomic fractions. **b** Top 15 most possible elemental combinations for ultraincompressible materials. The larger sphere size indicates materials with higher bulk moduli. Source data are provided as a Source Data file.

stiffness ($S_i$) can be derived as

$$S_i = Nkd^2/18. \tag{7}$$

The atomic stiffness ($S_i$) of diamond is calculated as 15.58 eV by substituting the parameters of chemical bonding ($N = 4$, $k = 467$ N m$^{-1}$, $d = 1.55$ Å[51]) into the relation ($S_i = Nkd^2/18$), which is consistent with the parameterized value from the product of bulk modulus and atomic volume (15.55 eV). It should be noted that since the bond force constant, bond length, and coordination number depend on local chemical environments, their parametrizations are challenging.

### Determination of atomic stiffnesses for some elements within the periodic table

For a few elements whose simple substances are gases, liquids, or weakly interacting molecular crystals at standard temperature and pressure (i.e., N, O, F, S, Cl, Br, and Hg), the atomic stiffnesses were derived from their compounds from the crystals in MP database that have a first-principles-calculated bulk modulus. We took oxygen as an example: Firstly, 1152 oxides with a first-principles-calculated bulk modulus in MP database were extracted. Then, we determined the atomic stiffness of oxygen (7.52 eV) by minimizing the difference between the predicted bulk moduli from our formula and first-

principles-calculated bulk moduli for these 1152 oxides. It should be noted that nitrogen is the only element among these elements that is possible to form high-bulk-modulus compounds (bulk modulus > 350 GPa). To guarantee the predictive accuracy preferentially for high-bulk-modulus materials, the atomic stiffness of nitrogen (16.32 eV) was derived from 20 nitrides with the first-principles-calculated bulk modulus higher than 350 GPa.

### Data cleaning of crystal structures

We did data cleaning of crystal structures (13172) that have a first-principles calculated bulk modulus extracted from Materials Project (MP). The criteria of data cleaning are as follows: Crystals (1) containing radioactive elements, (2) containing more than 20 atoms in the primitive cell, or (3) having an energy-above-hull higher than 1.0 eV atom$^{-1}$, or (4) having a nonpositive definite stiffness matrix or too low first-principles-calculated bulk modulus (lower than 30 GPa) were filtered out. Afterward, 6192 compounds were retained for evaluating the predictive performance of bulk moduli from atomic stiffnesses.

We also did data cleaning of crystal structures (1066695) whose bulk moduli remain unknown extracted from two of the largest databases, Materials Project (MP) and Open Quantum Materials Database (OQMD) before the high-throughput prediction of bulk moduli. The criteria of data cleaning are as follows: Crystals (1) containing

radioactive elements, (2) containing more than 20 atoms in the primitive cell, or (3) having an energy-above-hull higher than 1.0 eV atom[-1] were filtered out. Afterward, 512,419 compounds were retained for further high-throughput screening of ultraincompressible materials.

## Calculation of the mean absolute error and mean absolute relative error

To evaluate the accuracy of the predicted bulk moduli, the mean absolute relative errors (MAREs) and mean absolute errors (MAEs) of the predicted bulk moduli were calculated with respect to referred bulk moduli (first-principles-calculated or experimentally measured bulk moduli). Herein, the MAE and MARE were calculated by the following equations:

$$\text{MAE} = \frac{1}{N}\sum_{i=1}^{N}|B - B_R|, \tag{8}$$

$$\text{MARE} = \frac{1}{N}\sum_{i=1}^{N}\frac{|B - B_R|}{B_R} \times 100\%, \tag{9}$$

where $B$, $B_R$, and $N$ are the predicted bulk modulus, the referred bulk modulus, and the number of materials in the calculations, respectively.

## Generalized prediction from previous empirical/semiempirical formulae

Previous empirical/semiempirical formulae (Cohen's formula[14] and Li's formula[11]) are only applicable for accurate prediction of bulk moduli for a narrow range of crystals. To compare previous empirical/semiempirical formulae with our formula for a diverse set of crystals, we tried to generalize Cohen's formula[14] and Li's formula[11] for predicting 6192 compounds in MP database. For the generalized Cohen's formula[14], the bulk modulus for a crystal is calculated by

$$B = (N_c/4)(1972 - 220I)d^{-3.5}, \tag{10}$$

where $N_c$, $I$, and $d$ are the average coordination number, empirical ionicity parameter [$I = 2.9 + 0.6\ln(f_e)$, where $f_e$ is the ionicity of the crystal], and the average bond length of the crystal, respectively. For the generalized Li's formula[11], the bulk modulus for a crystal is calculated by

$$B = \frac{1944.8\sum w_{ab}x_{ab}/v + 13.5}{\exp[(1/\sum w_{ab}/f_{e(ab)})^2]}, \tag{11}$$

where $w_{ab}$, $x_{ab}$, $v$, and $f_{e(ab)}$ are the weighting factor of a-b bond, bond electronegativity, average bond volume, and effective ionicity, respectively.

## High-fidelity first-principles calculations

First-principles calculations were performed under the framework of the density functional theory by using the Vienna Ab initio Simulation Package[52]. The generalized gradient approximation of the Perdew-Burke-Ernzerhof functional[53] was used for the exchange and correlation interactions of electrons. Compared with first-principles calculations in the MP database, the following more stringent calculation options were adopted to ensure high-fidelity first-principles results. Considering the dispersion interactions, van der Waals corrections using the DFT-D3[54] method were adopted in all calculations. The spin-polarized calculations were adopted in relevant calculations. An energy cut-off of 1.3 times the default value was used, and a $k$-point mesh with a density of greater than 40 Å was used for Brillouin zone sampling. All structures were fully relaxed using a conjugate gradient algorithm with a stringent convergence criterion of the force on each atom ($10^{-3}$ eV Å$^{-1}$).

## Data availability

The data generated and analyzed in this study are available within the article and its Supplementary Information/Source Data file. The raw data used in the screening process are available at https://oqmd.org and https://materialsproject.org. Source data are provided with this paper.

## Code availability

The code used in this study is available from the corresponding author (E.G.) upon request.

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

## Acknowledgements

E.G., R.J., and X.Y. acknowledge support by the National Natural Science Foundation of China (12172261). R.J. acknowledges the technical assistance from Chunbo Zhang. The numerical calculations in this work have been performed on a supercomputing system in the Supercomputing Center of Wuhan University.

## Author contributions

E.G. conceived the idea and wrote the manuscript. R.J., X.Y., and E.G. did the calculations and analyses. R.J. and X.Y. contributed equally.

## Competing interests

The authors declare no competing interests.
