## [Peer Review File · Nature Communications]

Reviewer #1 (Remarks to the Author):

In this manuscript, the authors report a new method for fast screening of ultra-incompressible materials from database. In particular, a new concept, atomic stiffness, was introduced and calculated for most elements in the periodic table. Based on this, an analytic formula was derived for the calculation of bulk moduli of materials. According to the results reported in the manuscript and SI, the method proposed here shows advantages of high accuracy, efficiency, through a comparison with empirical/semiempirical formulae and even machine learning models. This research is quite interesting and might bring new ideas to scientists for new materials design. However, there are still some concerns that need to be addressed.

(1) It is believed that the resisting ability of a chemical bond to compression, which is the origin of the bulk modulus of crystals. Here the authors use the sum of atomic stiffness to calculate bulk modulus of materials. The chemical bonding is not considered. I am still not convinced at this point, although the calculated results seem to be good. The authors also mentioned that the bulk moduli for diamond and graphite are so different but atomic stiffness is almost same. Are there more theoretical explanations for this?

(2) How did the authors calculate the atomic stiffness? In the manuscript, "we determined the atomic stiffness for each element within the periodic table from the corresponding crystalline form (Fig. 2a)". I thought it was calculated from elemental crystals, which contain only one type of element. However, in the supporting information note 1, the authors mentioned that "we determined the atomic stiffness of oxygen (7.5 eV) by minimizing the difference between the predicted bulk moduli from our formula and first-principles calculated bulk moduli for these 1152 oxides." Could the authors make it clear about the calculation?

(3) How about the comparison of calculated values with experimental data of some known materials? It would be nice if the authors could survey the literature and give a comparison.

(4) The x-axis captions in Figure 2b and Figure 5b are missing.

(5) What is the meaning of sphere size and colors in Figure 5b?

(6) I could roughly guess how the authors calculated MARE and MAE. But it would be nice if calculation equations were given in the manuscript or SI.

Reviewer #2 (Remarks to the Author):

The authors derived and parameterized an analytic formula for the prediction of bulk moduli from the proposed "atomic stiffness". The derivation of this formula and the definition of atomic stiffness are sound. Based on this formula, the authors found 44 ultra-incompressible crystals with a bulk modulus rivaling diamond through high-throughput screening, showing great power of the prediction from atomic stiffnesses and its superiority over previous formulae and machine learning approach. I don't doubt the formula itself and its ability to make predictions, and think it's great that it gives researchers a new perspective on bulk modulus. However, a number of issues that arise in the article make its significance and novelty appear insufficient to be published in Nature Communications.

1. The author claim that "although different allotropes of the same element have very different bulk moduli and atomic volumes, their atomic stiffnesses are close". However, from Table S1, for element C, there is a large difference between MP-569416 with 17.48 and MP-24 with 13.56. Moreover, this claim contradicts to "we here used the dense crystalline form of simple substances to parametrize the atomic stiffness for each element" in Page 6. If there is not much difference, why choose the densest one?

2. The atomic stiffness of element C is 15.55 in Fig.2a. How was this value obtained, it seems that this does not correspond to diamond, because text in Page 4 states that diamond is 15.5.

3. In Fig.2b, why these parameters show periodic dependence upon the atomic number?

4. In Discussion section, the authors raised two questions that are very worth exploring. Unfortunately, the author does not answer these questions properly. From the 44 high-modulus materials discovered by the authors and the treasure map given, the design rule of ultra-incompressible materials is still consistent with the design principles that were established long ago, e.g., combination of low Z elements with transition metals. These are well-known and predictable, and the authors should give more profound design insights from their studies.

5. In Discussion section, the authors claim "our theoretical formula can be used for predicting the upper bound.....". Why is the upper bound, I do not see the physical meaning behind it. The reason obviously would not be that the dense phase was chosen in defining the atomic stiffness, because firstly, the authors claim that there is little difference between different polytypes, and secondly, that the upper bound for a single element does not strictly guarantee that the compound also corresponds upper bound.

6. The title of the article is not so appropriate because it does not seem to capture the innovative point of the article.

I cannot recommend this article for publication on Nature Communications until these issues are resolved.

Reviewer #3 (Remarks to the Author):

This paper merely calculates the multiplication of the atomic volume and bulk modulus from databases such as Materials Project, estimates the bulk modulus of many compounds with the molar fraction, plots them with the bulk modulus of the databases mimicking the machine learning process. What did they obtain from the machine learning? There are only two averaged scalar properties, the atomic volume and bulk modulus. In addition, the bulk modulus is referred from database; the adjustable parameter is only the atomic volume. There is no description about atom species, lattice structure, etc. in the learning data, so that the reviewer judged the authors didn't use machine learning. This paper contains such misleading descriptions so that it should be rejected. For example, the graph of the atomic stiffness in Fig.2b artificially resembles that of bulk modulus, despite of the remarkable peak in the graph of atomic volume; that is, it is clear that the lower graph in Fig.2b is NOT the multiplication of the upper and middle graphs, in spite of the definition of $S_i = B_i V_i$.

Point-by-point response to the reviewers' comments

Reviewer #1

In this manuscript, the authors report a new method for fast screening of ultra-incompressible materials from database. In particular, a new concept, atomic stiffness, was introduced and calculated for most elements in the periodic table. Based on this, an analytic formula was derived for the calculation of bulk moduli of materials. According to the results reported in the manuscript and SI, the method proposed here shows advantages of high accuracy, efficiency, through a comparison with empirical/semiempirical formulae and even machine learning models. This research is quite interesting and might bring new ideas to scientists for new materials design. However, there are still some concerns that need to be addressed.

Response: Thank you very much for your insightful comments and kind suggestions. We improved our work accordingly. Please find below our point-to-point responses (in blue) to your comments (in black).

(1) It is believed that the resisting ability of a chemical bond to compression, which is the origin of the bulk modulus of crystals. Here the authors use the sum of atomic stiffness to calculate bulk modulus of materials. The chemical bonding is not considered. I am still not convinced at this point, although the calculated results seem to be good. The authors also mentioned that the bulk moduli for diamond and graphite are so different but atomic stiffness is almost same. Are there more theoretical explanations for this?

Response: Thank you for this comment. We divided this comment into two parts so that our response can be more clearly understood.

First, regarding the comment on the consideration of chemical bonding: The parameters of chemical bonds are implicitly included in the analytic formula, since it can be demonstrated that the proposed atomic stiffness of each atom (S_i) is related to the coordination number (N), the bond force constant (k), and the bond length (d) of chemical bonds around the atom. Taking diamond as an example ($S_i = Nkd^2/18$, see **Supplementary Note 1 for details**), by substituting the parameters of chemical bonding ($N = 4$, $k = 467$ N/m, $d = 1.55$ Å¹) into the relation ($S_i = Nkd^2/18$), the calculated atomic stiffness (15.58 eV) is consistent with the parameterized value from the product of the bulk modulus and the atomic volume (15.55 eV). It should be noted that since the bond force constant, bond length, and coordination number depend on local chemical environments, their parametrizations are challenging. To address this issue, we circumvented these challenging parametrizations by using the new concept of atomic stiffness that is insensitive to local chemical environments.

Second, regarding the concern on the similarity of atomic stiffness values for diamond and graphite: We conducted a literature survey and analyses, providing two explanations accordingly: (1) It has been reported that the bulk modulus (B_i) has a relation with the bond length (d) as $B_i \sim d^n$, where n is approximately -3 ^{2,3}, while the atomic volume (V_i) has a relation with the bond length as $V_i \sim d^3$. Hence, the atomic stiffnesses ($S_i = B_i V_i$) of different allotropes for each element are almost constant. (2) It has been reported that the bulk modulus (B_i) is proportional to the volumetric cohesive energy ($\rho_i = E_i/V_i$, where E_i is the atomic cohesive energy): $B_i \sim E_i/V_i$ ⁴, *i.e.*, $B_i V_i = S_i \sim E_i$. As allotropes would be unstable if their energies are significantly higher than the most stable phase, the atomic cohesive energies of different allotropes (E_i) are usually very close (*e.g.*, 7.78 eV of diamond and 7.91 eV of graphite)⁵, resulting in similar atomic stiffnesses (S_i) for different allotropes of each element.

Accordingly, we added discussions on the chemical bonding and explanations on the insensitivities of atomic stiffness values on different allotropes into the revised manuscript. These discussions also help to clarify the physical concept of atomic stiffness.

Revisions made on page 3 of the main text:

From atomic stiffnesses, the bulk modulus can be predicted by

$$B = \sum x_i S_i / V. \quad (4)$$

It is known that the bulk modulus originates from the resisting ability of chemical bonds to compression. The parameters of chemical bonds are implicitly included in this analytic formula, since it has been demonstrated that the proposed atomic stiffness of each atom (S_i) is related to the coordination number (N), the bond force constant (k), and the bond length (d) of chemical bonds around the atom (see **Supplementary Note 1 for details**).

Revisions made on page 2 of the Supplementary Information:

Note 1: Correlation between the atomic stiffness and the chemical bonding parameters

Taking the diamond as an example, the atomic energy of an atom is contributed by the energy of the chemical bond around the atom:

$$E = NE_b/2, \quad (S1)$$

where N and E_b are the coordination number and the energy of a chemical bond, respectively. Meanwhile, E_b can be written as

$$E_b = kd^2 \varepsilon_b^2 / 2, \quad (S2)$$

where k , d , and ε_b are the bond force constant, the equilibrium bond length, and the bond strain, respectively. Herein, the bond strain (ε_b) has a relation with the volumetric strain (ε) under a small strain: $\varepsilon = 3\varepsilon_b$. From the definition of atomic stiffness ($S_i = d^2E/d\varepsilon^2|_{\varepsilon \rightarrow 0}$), the atomic stiffness (S_i) can be derived as

$$S_i = Nkd^2/18. \quad (S3)$$

The atomic stiffness (S_i) of diamond is calculated as 15.58 eV by substituting the parameters of chemical bonding ($N = 4$, $k = 467$ N/m, $d = 1.55 \text{ \AA}$) into the relation ($S_i = Nkd^2/18$), which is consistent with the parameterized value from the product of bulk modulus and atomic volume (15.55 eV). It should be noted that since the bond force constant, bond length, and coordination number depend on local chemical environments, their parametrizations are challenging. We circumvented these challenging parametrizations by using the new concept of atomic stiffness that is insensitive to local chemical environments.

Revisions made on page 4 of the main text:

The insensitivities of atomic stiffness values for different allotropes can be understood from the following analyses: (1) It has been reported that the bulk modulus (B_i) has a relation with the bond length (d) as $B_i \sim d^n$, where n is approximately -3 ^{2,3}, while the atomic volume (V_i) has a relation with the bond length as $V_i \sim d^3$. Hence, the atomic stiffnesses ($S_i = B_i/V_i$) of different allotropes for each element are almost constant. (2) It has been reported that the bulk modulus (B_i) is proportional to the cohesive energy density ($\rho_i = E_i/V_i$, where E_i is the atomic cohesive energy): $B_i \sim E_i/V_i$ ⁴, i.e., $B_i/V_i = S_i \sim E_i$. Because allotropes would be unstable if their energies are significantly higher than the most stable phase, the atomic cohesive energies of different allotropes (E_i) are usually very close (e.g., 7.78 eV of diamond and 7.91 eV of graphite)⁵, resulting in similar atomic stiffnesses (S_i) for different allotropes of each element.

(2) How did the authors calculate the atomic stiffness? In the manuscript, “we determined the atomic stiffness for each element within the periodic table from the corresponding crystalline form (Fig. 2a).”. I thought it was calculated from elemental crystals, which contain only one type of element. However, in the supporting information note 1, the authors mentioned that “we determined the atomic stiffness of oxygen (7.5 eV) by minimizing the difference between the predicted bulk moduli from our formula and first-principles calculated bulk moduli for these 1152 oxides.” Could the authors make it clear about the calculation?

Response: We are sorry for causing this confusion. The determination of atomic stiffness for each element within the periodic table is based on the following rules: (1) For most elements whose simple substances are solids at standard temperature and pressure, the atomic stiffness for each element was calculated from the product of the

bulk modulus and the atomic volume of the corresponding crystalline form. (2) For a few elements whose simple substances are gases or liquids at standard temperature and pressure (*i.e.*, N and O), the atomic stiffness for each element was derived from their compounds in MP database that have first-principles calculated bulk moduli (see **Supplementary Note 1 for details**). In response, we revised the statement in the manuscript to make it clear.

Revisions made on page 4 of the main text:

To predict bulk moduli from the atomic stiffnesses (**Eq. 4**), we determined the atomic stiffness (S_i) for each element within the periodic table (**Fig. 2a**).

Revisions made on page 5 of the main text:

The determination of atomic stiffness for each element within the periodic table is based on the following rules:

(1) For most elements whose simple substances are solids at standard temperature and pressure, we used the dense crystalline form of simple substances to parametrize the atomic stiffness by considering that the state of each element in the ultra-incompressible compounds is close to their dense crystalline form. The dense crystalline forms of simple substances were extracted from the MP database⁶. The atomic stiffness (S_i) for each element was calculated from the bulk modulus (B_i) and atomic volume (V_i) of the corresponding structure. (2) For a few elements whose simple substances are gases, liquids, or weakly interacting molecular crystals at standard temperature and pressure (*i.e.*, N and O), the atomic stiffness (S_i) for each element was derived from their compounds in MP database that have first-principles calculated bulk moduli (see **Supplementary Note 1 for details**).

(3) How about the comparison of calculated values with experimental data of some known materials? It would be nice if the authors could survey the literature and give a comparison.

Response: Thank you for this very valuable suggestion. In response, we did a literature survey and provided a comparison of calculated values from our formula with experimental data of known materials (**Fig. S3**). We added these useful results in the revised manuscript.

Revisions made on page 9 of the main text:

Finally, we did a literature survey and collected materials with experimentally measured bulk moduli⁷⁻²³ (**Table S4**). The predicted bulk moduli of these materials by our formula are well consistent with experimental values (**Fig. 3d**), which provides experimental support for our theoretical prediction.

Fig. 3|Comparison of bulk moduli predicted from our formula (B_O), previous empirical/semiempirical formulae (B_{Cohen} and B_{Li}) with first-principles calculations (B_F), and comparison of B_O and experimental measurements (B_{EXP}). Predicted bulk moduli of 6192 crystals in MP database from **a, our formula, **b**, Cohen's formula, and **c**, Li's formula, as compared with first-principles calculations from MP database (B_F). **d**, Comparison of B_O and B_{EXP} . Each point of mean absolute relative error (MARE) and mean absolute error (MAE) is counted for crystals within the range of ± 50 GPa.**

Revisions made on page 14 of the Supplementary Information:

Table S4. Comparison of bulk moduli from our formula (B_O) and experimental measurements (B_{EXP})⁷⁻²³.

Crystal	Atomic volume (\AA^3)	B_O (GPa)	B_{EXP} (GPa)
Diamond	5.71	436	442 ⁷
BN	5.96	348	369 ⁸
OsN ₂	9.11	402	358 ⁹
PtN ₂	9.24	329	354 ¹⁰
MoN	10.14	334	345 ¹¹
TaC	11.05	273	344 ¹²
WC	10.38	357	329 ¹²
Mo ₂ N	12.13	301	301 ¹¹
NbC	11.16	354	300 ¹²
RuO ₂	10.45	252	270 ¹³
HfC	12.48	197	242 ¹²
TiC	10.16	219	242 ¹⁴
β -SiN ₄	10.39	214	234 ¹⁵
ZrSiO ₄	9.72	183	230 ¹⁶
Cr ₂ O ₃	9.64	202	224 ¹⁷
ZrC	13.10	179	223 ¹⁴
SiC	10.50	199	211 ¹⁸

AlN	10.63	187	201 ¹⁹
GaN	11.74	154	180 ¹⁹
BP	11.75	136	173 ²⁰
MgO	9.64	106	156 ²¹
InN	16.13	114	139 ¹⁹
CaO	14.17	68	106 ¹⁷
BeS	14.48	61	105 ²²
Si	20.44	83	100 ²³

(4) The x-axis captions in Figure 2b and Figure 5b are missing.

Response: Thank you for pointing out these issues. In response, we added the x-axis captions in **Figure 2b** and **Figure 5b**.

a

H 1												He 2					
Li 3	Be 4											B 5	C 6	N 7	O 8	F 9	Ne 10
14	122											197	436	401	140	71	
20.12	7.89											7.78	5.71	6.51	8.60	9.31	
1.76	6.02											9.58	15.55	16.32	7.52	4.11	
Na 11	Mg 12											Al 13	Si 14	P 15	S 16	Cl 17	Ar 18
8	37											83	83	114	52	41	
36.32	22.75											16.47	20.44	14.66	15.60	20.92	
1.82	5.26											8.54	10.61	10.45	5.04	5.36	
K 19	Ca 20	Sc 21	Ti 22	V 23	Cr 24	Mn 25	Fe 26	Co 27	Ni 28	Cu 29	Zn 30	Ga 31	Ge 32	As 33	Se 34	Br 35	Kr 36
4	17	52	113	179	259	230	182	212	198	145	75	50	59	51	26	16	
72.85	41.76	24.69	17.36	13.40	11.87	10.52	11.45	10.92	10.77	11.87	15.56	20.16	23.92	22.45	25.84	26.49	
1.82	4.44	8.03	12.26	14.99	19.21	19.71	13.03	14.47	13.33	10.76	7.29	6.30	8.82	7.16	4.15	2.64	
Rb 37	Sr 38	Y 39	Zr 40	Nb 41	Mo 42	Tc 43	Ru 44	Rh 45	Pd 46	Ag 47	Cd 48	In 49	Sn 50	Sb 51	Te 52	I 53	Xe 54
3	11	41	94	174	262	300	308	253	192	88	45	40	39	44	33	15	
89.90	54.61	32.85	23.50	18.31	15.89	14.59	13.95	14.20	14.66	18.00	23.26	26.50	36.11	32.13	33.35	41.68	
1.69	3.75	8.42	13.81	19.91	26.02	27.36	26.86	22.45	17.64	9.90	6.54	6.70	8.80	8.93	6.81	3.88	
Cs 55	Ba 56	La-Lu	Hf 72	Ta 73	W 74	Re 75	Os 76	Ir 77	Pt 78	Au 79	Hg 80	Tl 81	Pb 82	Bi 83	Po 84	At 85	Rn 86
2	9		108	194	304	365	402	346	247	137	64	33	44	41			
114.05	63.64		22.48	18.34	16.19	15.06	14.36	14.56	15.72	18.15	24.67	29.80	30.85	35.36			
1.43	3.58		15.18	22.23	30.76	34.36	36.07	31.48	24.27	15.54	9.85	6.06	7.45	8.96			

Legend:
 — Atomic number
 — Bulk modulus (GPa)
 — Atomic volume (\AA^3)
 — Atomic stiffness (eV)

Nonmetal (green)
 Metal (blue)
 Transition metal (red)

Fig. 2|Atomic parameters for elements within the periodic table. a, Bulk moduli, atomic volumes, and atomic stiffnesses for elements within the periodic table. b, Periodic dependences of bulk moduli, atomic volumes, and atomic stiffnesses upon the atomic number.

Fig. 5|Treasure maps of elemental combinations for ultra-incompressible materials. a, Estimation of the bulk moduli for binary compounds of different elemental combinations. **b,** Top 15 most possible elemental combinations for ultra-incompressible materials. The larger sphere size and the darker color indicate materials with higher bulk moduli. The elemental combinations labeled in red have been also found in the ultra-incompressible materials as summarized in **Table 1**.

(5) What is the meaning of sphere size and colors in Figure 5b?

Response: Thank you for this comment. The larger sphere size and the darker color indicate materials with higher bulk moduli. We added a description of the meaning of sphere size and colors in **Figure 5b**.

See changes made to the caption of Fig. 5b:

The larger sphere size and the darker color indicate materials with higher bulk moduli.

(6) I could roughly guess how the authors calculated MARE and MAE. But it would be nice if calculation equations were given in the manuscript or SI.

Response: Thank you for this useful suggestion. In response, we provided the calculation equations and details of MARE and MAE in the method section of the manuscript.

Revisions made on page 13 of the main text:

Calculation of the mean absolute error and mean absolute relative error. To evaluate the accuracy of the predicted bulk moduli, the mean absolute relative errors (MAREs) and mean absolute errors (MAEs) of the predicted bulk moduli were calculated with respect to the referred bulk moduli (first-principles-calculated or experimentally measured bulk moduli). Herein, the MAE and MARE were calculated by the following equations:

$$\text{MAE} = \frac{1}{N} \sum_{i=1}^N |B - B_R|, \quad (5)$$

$$\text{MARE} = \frac{1}{N} \sum_{i=1}^N \frac{|B - B_R|}{B_R} \times 100\%, \quad (6)$$

where B , B_R , and N are the predicted bulk modulus, the referred bulk modulus, and the number of materials using in the calculations, respectively.

Reviewer #2

The authors derived and parameterized an analytic formula for the prediction of bulk moduli from the proposed “atomic stiffness”. The derivation of this formula and the definition of atomic stiffness are sound. Based on this formula, the authors found 44 ultra-incompressible crystals with a bulk modulus rivaling diamond through high-throughput screening, showing great power of the prediction from atomic stiffnesses and its superiority over previous formulae and machine learning approach. I don't doubt the formula itself and its ability to make predictions, and think it's great that it gives researchers a new perspective on bulk modulus. However, a number of issues that arise in the article make its significance and novelty appear insufficient to be published in Nature Communications.

Response: We appreciate your insightful comments and kind suggestions. We improved our work accordingly. Please find below our point-to-point responses (in blue) to your comments (in black). The revisions are shown in blue color in the revised manuscript.

1. The author claim that “although different allotropes of the same element have very different bulk moduli and atomic volumes, their atomic stiffnesses are close”. However, from Table S1, for element C, there is a large difference between MP-569416 with 17.48 and MP-24 with 13.56. Moreover, this claim contradicts to “we here used the dense crystalline form of simple substances to parametrize the atomic stiffness for each element” in Page 6. If there is not much difference, why choose the densest one?

Response: Thank you for this valuable comment. We have divided this comment into two parts so that our response can be more clearly understood.

First, for the comment on the difference between MP-569416 and MP-24: Upon closer examination, we found that MP-569416 represents graphite with an unusually large interlayer distance exceeding 7 Å, which is a result of calculation issues in the Materials Project, such as the absence of van der Waals corrections in high-throughput DFT calculations. By employing high-fidelity first-principles calculations for structural optimization, the atomic stiffness of MP-569416 matched that of graphite and was similar to other carbon allotropes. To address this thoroughly, we reevaluated and recalculated different allotropes using high-fidelity first-principles calculations, and **Table S1** was polished accordingly. The mean absolute relative error (MARE) for atomic stiffness parametrized from different allotropes is within 11%.

Second, regarding the choice of dense crystalline forms for parametrizing atomic stiffness: As you noted, choosing the dense crystalline form has a minimal impact on the parametrized atomic stiffness. For instance, we compared

the predicted bulk moduli from first-principles calculations (B_F) and our formula (B_O), where the atomic stiffness is parametrized from the dense crystalline form and the most stable form of simple substances, respectively (Fig. R1). These findings suggest that selecting the dense crystalline form for simple substances has little influence on bulk moduli predictions. This is due to the minimal difference in atomic stiffness when parametrized from different allotropes (Table S1). We chose the dense crystalline form because the state of each element in ultra-incompressible compounds is similar to their dense crystalline form of simple substances, and the dense crystalline form of simple substances usually has good stabilities at standard temperature and pressure. Notably, in most cases, the dense crystalline form is also the most stable form. However, for a few elements—B, C, Cs, K, Rb, and Sr—the dense crystalline form is not the most stable form, but its energy is very close to the most stable form. For example, we chose diamond, the dense crystalline form of carbon, whose energy is very close to that of graphite, the most stable form.

Fig. R1|Comparison of bulk moduli predicted from our formula (B_O) with first-principles calculations (B_F). **a**, The atomic stiffness is parameterized from the dense crystalline form of of simple substances. **b**, The atomic stiffness is parameterized from the most stable form of simple substances.

Revisions made on page 8 of the Supplementary Information:

Table S1. Atomic stiffnesses of different allotropes for each element within the periodic table. The mean absolute relative error (MARE) is calculated with respect to the corresponding dense crystalline form.

MP-ID	Elements	S (eV)	MARE	MP-ID	Elements	S (eV)	MARE
MP-8566	Ag	9.72	0.01	MP-110	Mg	5.16	0.02
MP-10597	Ag	9.96		MP-973364	Mg	5.16	
MP-124	Ag	9.90		MP-153	Mg	5.27	
MP-134	Al	8.54	0.00	MP-129	Mo	26.02	0.00
MP-11	As	7.16	0.00	MP-982370	Na	2.03	0.08
MP-81	Au	15.54	0.00	MP-567772	Na	1.75	
MP-160	B	9.61	0.00	MP-127	Na	1.82	
MP-161	B	9.58		MP-75	Nb	19.91	0.00
MP-56	Ba	3.34	0.07	MP-10257	Ni	13.28	0.00
MP-122	Ba	3.58		MP-1008728	Ni	13.37	
MP-20	Be	6.06	0.01	MP-23	Ni	13.33	
MP-87	Be	6.02		MP-8643	Os	36.76	0.02
MP-23157	Bi	8.93	0.00	MP-49	Os	36.07	
MP-23152	Bi	8.96		MP-53	P	10.45	0.00
MP-24	C	13.56	0.04	MP-20745	Pb	8.39	0.01
MP-570002	C	14.09		MP-20483	Pb	8.49	
MP-1008374	C	14.51		MP-2	Pd	17.64	0.00
MP-569416	C	15.09		MP-126	Pt	24.27	0.00
MP-48	C	15.09		MP-604321	Rb	1.72	0.02
MP-1008395	C	15.18		MP-867126	Rb	1.71	
MP-169	C	15.25		MP-70	Rb	1.69	
MP-568286	C	15.28		MP-8642	Re	34.29	0.00
MP-1095534	C	15.28		MP-975065	Re	34.31	
MP-47	C	15.58		MP-8	Re	34.36	
MP-611426	C	15.56		MP-74	Rh	22.45	0.00
MP-616440	C	15.55		MP-8639	Ru	27.03	0.01
MP-66	C	15.55		MP-33	Ru	26.86	
MP-21	Ca	4.33	0.03	MP-104	Sb	8.93	0.00
MP-132	Ca	4.17		MP-36	Sc	7.85	0.02
MP-166	Ca	4.49		MP-67	Sc	8.03	
MP-45	Ca	4.44		MP-10649	Si	8.42	0.11
MP-669382	Co	14.01	0.02	MP-1014212	Si	8.68	
MP-102	Co	14.46		MP-676001	Si	8.45	
MP-54	Co	14.47		MP-644693	Si	8.71	
MP-17	Cr	18.36	0.04	MP-571520	Si	9.13	
MP-90	Cr	19.21		MP-16220	Si	9.12	
MP-639727	Cs	1.44	0.01	MP-109	Si	9.67	
MP-1	Cs	1.43		MP-168	Si	9.56	
MP-989695	Cu	10.49	0.03	MP-165	Si	11.37	

MP-30	Cu	10.76		MP-34	Si	10.26	
MP-150	Fe	13.10	0.01	MP-999200	Si	11.07	
MP-13	Fe	13.03		MP-971661	Si	11.06	
MP-569423	Ga	5.76	0.08	MP-971662	Si	11.05	
MP-10021	Ga	5.82		MP-92	Si	10.35	
MP-567540	Ga	5.90		MP-149	Si	10.61	
MP-142	Ga	6.30		MP-867202	Sr	4.11	0.07
MP-1008733	Ge	8.29	0.09	MP-76	Sr	4.10	
MP-998883	Ge	8.13		MP-95	Sr	4.03	
MP-128	Ge	7.28		MP-139	Sr	3.80	
MP-137	Ge	7.26		MP-42	Ta	22.81	0.03
MP-78	Ge	8.24		MP-569794	Ta	22.81	
MP-1007760	Ge	8.75		MP-50	Ta	22.23	
MP-32	Ge	8.82		MP-8638	Tc	29.19	0.03
MP-8640	Hf	14.20	0.06	MP-867351	Tc	27.57	
MP-103	Hf	15.18		MP-113	Tc	27.36	
MP-10157	K	1.84	0.01	MP-567313	Te	6.81	0.00
MP-972981	K	1.83		MP-19	Te	6.81	
MP-604318	K	1.83		MP-6985	Ti	11.60	0.03
MP-58	K	1.82		MP-72	Ti	12.08	
MP-1018134	Li	1.90	0.02	MP-46	Ti	12.26	
MP-10173	Li	1.77		MP-82	Tl	6.06	0.00
MP-567337	Li	1.77		MP-146	V	14.99	0.00
MP-976411	Li	1.76		MP-11334	W	30.78	0.00
MP-1063005	Li	1.76		MP-91	W	30.76	
MP-51	Li	1.76		MP-9	Y	7.91	0.06
MP-135	Li	1.76		MP-112	Y	8.42	
				MP-1077723	Zr	13.45	0.03
				MP-131	Zr	13.81	

Note: The data in this table is sourced from the Materials Project, which has some known calculation issues in its high-throughput DFT calculations, such as the lack of van der Waals corrections. To address these concerns, we conducted a thorough examination of the structures using high-fidelity first-principles calculations and *ab initio* molecular dynamics (AIMD) simulations. We excluded any structures from Table S1 that were mechanically unstable (i.e., having nonpositive definite stiffness matrices) or unable to maintain structural stability at 300 K for 10 ps.

2. The atomic stiffness of element C is 15.55 in Fig. 2a. How was this value obtained, it seems that this does not correspond to diamond, because text in Page 4 states that diamond is 15.5.

Response: We apologize for the confusion resulting from rounding off decimals. The more accurate atomic stiffness of element C, corresponding to diamond, is 15.5466 eV. When rounded, this value becomes 15.5 eV to one decimal place and 15.55 eV to two decimal places. To address this issue, we have updated the manuscript to present atomic stiffness values rounded to two decimal places. Furthermore, we have carefully reviewed and polished the entire manuscript to maintain consistency.

Revisions made on page 4 of the main text:

For example, although the bulk moduli for diamond (436 GPa) and graphite (237 GPa) differ significantly, the atomic stiffnesses for diamond (15.55 eV) and graphite (15.09 eV) only have a difference of less than 3%.

3. In Fig. 2b, why these parameters show periodic dependence upon the atomic number?

Response: Thank you for this comment. It has been reported that the elements within the same group in the periodic table exhibit similar atomic properties, such as atomic volume and atomic valance electron, which show periodic dependence upon the atomic number²⁴. Moreover, the bulk moduli also exhibit periodic dependence upon the atomic number, since it is related to the valance electron density, *i.e.*, atomic valance electron divided by atomic volume²⁴. Consequently, as the product of bulk modulus and atomic volume, the atomic stiffness also shows periodic dependence upon the atomic number. In response, we added an explanation of this periodic dependence in the revised manuscript.

Revisions made on page 5 of the main text:

These bulk moduli, atomic volumes, and atomic stiffnesses (S_i) exhibit periodic dependences upon the atomic number (**Fig. 2b**). This is because the elements in the same group in the periodic table have similar atomic properties, such as atomic volume, atomic valance electron, which shows periodic dependence upon the atomic number²⁴. Moreover, the bulk moduli also exhibit periodic dependence upon the atomic number, since it is related to the valance electron density, *i.e.*, atomic valance electron divided by atomic volume²⁴. Consequently, as the product of bulk modulus and atomic volume, the atomic stiffness also shows periodic dependence upon the atomic number.

4. In Discussion section, the authors raised two questions that are very worth exploring. Unfortunately, the author does not answer these questions properly. From the 44 high-modulus materials discovered by the authors and the treasure map given, the design rule of ultra-incompressible materials is still consistent with the design principles

that were established long ago, e.g., combination of low Z elements with transition metals. These are well-known and predictable, and the authors should give more profound design insights from their studies.

Response: We appreciate your valuable feedback and suggestion. In response, we provided further design insights from our studies by contrasting our study with previously reported design principles.

Prior design principles suggest that potential ultra-incompressible materials include: (1) compounds composed of light elements (B, C, N, O, etc.) from periods 2 and 3 of the periodic table, capable of forming short covalent bonds, and (2) compounds combining heavy transition metal elements with light elements, as heavy transition metals contribute high valence electron density to the compounds, thus improving resistance to mechanical deformation²⁵⁻²⁷. It should be noted that these prior design principles are mainly empirical and qualitative, while our formula offers a theoretical and quantitative approach (**Fig. 5**). Consequently, our studies provide much more accurate predictions and insights for designing ultra-incompressible materials (bulk modulus > 400 GPa) compared to previous design principles:

1. The range of elements potentially forming ultra-incompressible materials is significantly narrowed. For instance, among light elements, carbon (C) and nitrogen (N) are the most promising candidates, while boron (B) and oxygen (O) are unlikely to form ultra-incompressible materials (bulk modulus > 400 GPa).
2. Alloys composed of heavy transition metal elements represent a new class of materials with the potential for ultra-incompressibility.
3. Once the elemental composition is determined for ultra-incompressible materials, the focus should shift to identifying structures with small atomic volumes (as indicated by **Eq. 4**).

These treasure maps and insights provide guidelines for designing and discovering ultra-incompressible materials of the future.

Revisions made on page 11 of the main text:

More specifically, previous design principles indicate that potential super-hard, ultra-incompressible materials include (1) compounds composed of light elements (B, C, N, O, etc.) from periods 2 and 3 of the periodic table, as these elements can form short covalent bonds, and (2) compounds combining heavy transition metal elements with light elements, since the heavy transition metals contribute high valence electron density to the compounds, which enhances their resistance to mechanical deformation²⁵⁻²⁷. It should be noted that these previous design principles are mainly empirical and qualitative, while our formula offers a theoretical and quantitative approach

(Fig. 5). As a result, our studies provide more accurate predictions and insights for designing ultra-incompressible materials (bulk modulus > 400 GPa) compared to previous design principles: (1) The range of elements that can potentially form ultra-incompressible materials is significantly narrowed. For instance, among light elements, carbon (C) and nitrogen (N) are the most promising candidates, while boron (B) and oxygen (O) are unlikely to form ultra-incompressible materials (bulk modulus > 400 GPa). (2) Alloys composed of heavy transition metal elements are a new family of materials with the potential to be ultra-incompressible. (3) Once the elemental composition is determined in the search for ultra-incompressible materials, efforts should be directed towards identifying structures with small atomic volumes (as indicated by Eq. 4). These treasure maps and insights provide guidelines for designing and discovering ultra-incompressible materials of the future.

5. In Discussion section, the authors claim “our theoretical formula can be used for predicting the upper bound.....”. Why is the upper bound, I do not see the physical meaning behind it. The reason obviously would not be that the dense phase was chosen in defining the atomic stiffness, because firstly, the authors claim that there is little difference between different polytypes, and secondly, that the upper bound for a single element does not strictly guarantee that the compound also corresponds upper bound.

Response: Thank you for your insightful comment. Our previous claim that the predicted bulk modulus is an upper-bound estimation was due to the assumption that the overall volume of ultra-incompressible materials is approximately equal to the sum of the atomic volume of each atom, where the atomic volume of each atom is parameterized from the dense crystalline form of simple substances. Considering that the overall volume is likely to be underestimated due to this assumption, the predicted bulk modulus is likely to be overestimated according to Eq. 4.

To address your concern, we analyzed the validity of this assumption. Interestingly, Fig. S4 demonstrates that the sum of the atomic volume of each atom parameterized from the dense crystalline form of simple substances aligns well with the actual total volume of compounds. These findings indicate that the assumption is valid, and the predicted bulk modulus is not overestimated due to the assumption. Consequently, we have revised the related claim concerning the upper-bound estimation.

Revisions made on page 11 of the main text:

Considering that the overall volume of ultra-incompressible materials approximately equals the sum of the atomic volume of each atom therein, our theoretical formula can be used for predicting the bulk moduli of various elemental combinations by substituting the determined atomic stiffness for each element within the periodic table.

Revisions made on page 7 of the Supplementary Information:

Fig. S3. Comparison of the sum of atomic volume and the actual total volume of compounds.

6. The title of the article is not so appropriate because it does not seem to capture the innovative point of the article.

Response: Thank you for this insightful reminder. We believe that the innovative point recognized by the reviewer is the powerful prediction of bulk moduli from atomic stiffnesses. To capture this innovative point of the article, we revised the title to “High-throughput screening of ultra-incompressible materials directed by powerful prediction of bulk moduli from atomic stiffnesses”.

I cannot recommend this article for publication on Nature Communications until these issues are resolved.

Response: We do appreciate the reviewer for insightful and constructive comments. Accordingly, we tried our best to improve the quality of our manuscript. We hope that these improvements have resolved these issues.

Reviewer #3

This paper merely calculates the multiplication of the atomic volume and bulk modulus from databases such as Materials Project, estimates the bulk modulus of many compounds with the molar fraction, plots them with the bulk modulus of the databases mimicking the machine learning process. What did they obtain from the machine learning? There are only two averaged scalar properties, the atomic volume and bulk modulus. In addition, the bulk modulus is referred from database; the adjustable parameter is only the atomic volume. There is no description about atom species, lattice structure, etc. in the learning data, so that the reviewer judged the authors didn't use machine learning.

Response: Thank you for this comment. We would like to clarify that our study does not use the machine learning approach. Instead, we propose an analytic formula that demonstrates clear superiorities of high accuracy, high efficiency, strong universality, and high interpretability compared to other methods, including machine learning. As this reviewer noted, our formula does not depend on more factors such as atom species or lattice structure, which are usually used in previous methods (empirical/semiempirical formulae and machine learning models). The limited number of adjustable parameters in our formula contributes to its simplicity and allows for powerful predictions of bulk moduli, which is a major innovation. This simplicity is achieved by introducing the concept of atomic stiffness that is insensitive to local chemical environments for each element. This theoretical derivation avoids the challenge to parameterize other complex factors that are sensitive to local chemical environments for each element, such as bond force constant, bond length. To address the reviewer's concerns, we have demonstrated how this formula's simplicity is achieved by additionally exploring the correlation between atomic stiffness and chemical bonding parameters, and have provided explanations on the insensitivities of atomic stiffness values across different allotropes in the revised manuscript. We believe that these revisions will help to clarify the physical concept of atomic stiffness and emphasize the value of our work.

Revisions made on page 2 of the Supplementary Information:

Note 1: Correlation between the atomic stiffness and the chemical bonding parameters

Taking the diamond as an example, the atomic energy of an atom is contributed by the energy of the chemical bond around the atom:

$$E = NE_b/2, \tag{S1}$$

where N and E_b are the coordination number and the energy of a chemical bond, respectively. Meanwhile, E_b can be written as

$$E_b = kd^2\varepsilon_b^2/2, \quad (S2)$$

where k , d , and ε_b are the bond force constant, the equilibrium bond length, and the bond strain, respectively. Herein, the bond strain (ε_b) has a relation with the volumetric strain (ε) under a small strain: $\varepsilon = 3\varepsilon_b$. From the definition of atomic stiffness ($S_i = d^2E/d\varepsilon^2|_{\varepsilon \rightarrow 0}$), the atomic stiffness (S_i) can be derived as

$$S_i = Nkd^2/18. \quad (S3)$$

The atomic stiffness (S_i) of diamond is calculated as 15.58 eV by substituting the parameters of chemical bonding ($N = 4$, $k = 467$ N/m, $d = 1.55 \text{ \AA}$) into the relation ($S_i = Nkd^2/18$), which is consistent with the parameterized value from the product of bulk modulus and atomic volume (15.55 eV). It should be noted that since the bond force constant, bond length, and coordination number depend on local chemical environments, their parametrizations are challenging. We circumvented these challenging parametrizations by using the new concept of atomic stiffness that is insensitive to local chemical environments.

Revisions made on page 4 of the main text:

The insensitivities of atomic stiffness values for different allotropes can be understood from the following analyses: (1) It has been reported that the bulk modulus (B_i) has a relation with the bond length (d) as $B_i \sim d^n$, where n is approximately -3 ^{2,3}, while the atomic volume (V_i) has a relation with the bond length as $V_i \sim d^3$. Hence, the atomic stiffnesses ($S_i = B_iV_i$) of different allotropes for each element are almost constant. (2) It has been reported that the bulk modulus (B_i) is proportional to the cohesive energy density ($\rho_i = E_i/V_i$, where E_i is the atomic cohesive energy): $B_i \sim E_i/V_i$ ⁴, *i.e.*, $B_iV_i = S_i \sim E_i$. Because allotropes would be unstable if their energies are significantly higher than the most stable phase, the atomic cohesive energies of different allotropes (E_i) are usually very close (*e.g.*, 7.78 eV of diamond and 7.91 eV of graphite)⁵, resulting in similar atomic stiffnesses (S_i) for different allotropes of each element.

This paper contains such misleading descriptions so that it should be rejected. For example, the graph of the atomic stiffness in Fig.2b artificially resembles that of bulk modulus, despite of the remarkable peak in the graph of atomic volume; that is, it is clear that the lower graph in Fig.2b is NOT the multiplication of the upper and middle graphs, in spite of the definition of $S_i=B_iV_i$.

Response: We have thoroughly reviewed the raw data and can confirm that the lower graph is indeed the product of the upper and middle graphs. It should be noted that the raw data that supports our claim have been provided in Fig. 2a. We hope our responses and revisions adequately address the reviewer's concerns and offer a better understanding of our work.

Reviewer #1 (Remarks to the Author):

My concerns have been addressed in the revised manuscript. Now the manuscript is clear for me. I have no further comments and questions.

Reviewer #2 (Remarks to the Author):

The author's have answered my initial comments quite well. I recommend its publication and I think that this manuscript will be of interest to the readers of Nature Communications.